# Multiscale Geographically Weighted Regression in the Investigation of Local COVID-19 Anomalies Based on Population Age Structure in Poland

**DOI:** 10.3390/ijerph20105875

**Published:** 2023-05-19

**Authors:** Mateusz Ciski, Krzysztof Rząsa

**Affiliations:** Faculty of Geoengineering, Institute of Spatial Management and Geography, Department of Socio-Economic Geography, University of Warmia and Mazury in Olsztyn, 10-720 Olsztyn, Poland; krzysztof.rzasa@uwm.edu.pl

**Keywords:** COVID-19, SARS-CoV-2, pandemic, multiscale geographically weighted regression, MGWR, geographic information system, GIS

## Abstract

A growing number of various studies focusing on different aspects of the COVID-19 pandemic are emerging as the pandemic continues. Three variables that are most commonly used to describe the course of the COVID-19 pandemic worldwide are the number of confirmed SARS-CoV-2 cases, the number of confirmed COVID-19 deaths, and the number of COVID-19 vaccine doses administered. In this paper, using the multiscale geographically weighted regression, an analysis of the interrelationships between the number of confirmed SARS-CoV-2 cases, the number of confirmed COVID-19 deaths, and the number of COVID-19 vaccine doses administered were conducted. Furthermore, using maps of the local R^2^ estimates, it was possible to visualize how the relations between the explanatory variables and the dependent variables vary across the study area. Thus, analysis of the influence of demographic factors described by the age structure and gender breakdown of the population over the course of the COVID-19 pandemic was performed. This allowed the identification of local anomalies in the course of the COVID-19 pandemic. Analyses were carried out for the area of Poland. The results obtained may be useful for local authorities in developing strategies to further counter the pandemic.

## 1. Introduction

The COVID-19 pandemic became one of the major health challenges of the 21st century and has changed the world today. The pandemic has inflicted almost unimaginable harm on the life, health, and economy of many nations [1]. From the beginning, it has also become the subject of different scientific research.

The available scientific databases are particularly full of studies on the impact of the COVID-19 pandemic on human health [2,3,4,5,6,7,8,9,10,11]. It was proven to have a negative impact on both physical and mental health [12,13,14,15,16,17]. For example, during the initial phase of the COVID-19 outbreak in China, more than half of the respondents rated their psychological impact as moderate-to-severe, and about one-third reported moderate-to-severe anxiety [2]. Having an acquaintance infected with COVID-19 increased both depression and stress, whereas a history of stressful situations and medical problems raised depression and anxiety levels [18]. Negative changes in health behaviors are associated with increased psychological distress during the COVID-19 pandemic [17,19]. The pandemic has also clearly affected people’s physical activity [20,21]. Public health restrictions impacted the physical activity of the elderly, especially those who had higher levels of sports activity before the pandemic [22].

The issue of the COVID-19 vaccine has become a basis of the research carried out by various scientists [23,24,25,26,27,28,29]. Vaccination, considered one of the most effective public health measures, has always had a significant impact on the fight against various infectious diseases. COVID-19 vaccines were found to have an effect on reducing the risk of ‘Long COVID’ [23]. Some studies provide valuable information regarding the COVID-19 vaccine booster hesitancy and the potential variables influencing it [30], as well as factors associated with overall COVID-19 vaccine hesitancy [1,31]. No associations were found between willingness to vaccinate and age; however, females were more likely to be unsure than unwilling to vaccinate against COVID-19 compared to males [32]. Acceptance of the COVID-19 vaccine may play a major role in combating the pandemic [5]. Acceptance and willingness to receive the vaccine are among the main factors in the success or failure of a health system in implementing the vaccination program [33]. Wide variability in COVID-19 vaccine acceptance rates has been reported in different countries and regions of the world. A significant number of studies reported COVID-19 acceptance rates below 60%, which would pose a serious problem for efforts to control the current COVID-19 pandemic [34]. According to the research, the most adequate strategy to enhance immunization and ensure that it is widely accepted is to promote health educational interventions for professionals working in the primary health care sector [35]. The availability of an effective vaccine and wide coverage are two crucial factors for the success of vaccination programs [36].

One consequence of the COVID-19 pandemic was also an increase in mortality [37,38]. Specific comorbidities, mainly of cardiovascular nature, and medications at the time of infection could explain around one quarter of the mortality from the COVID-19 disease [39]. A mild correlation was found of mortality and morbidity rates with the population density and the percentage of the elderly population [40]. The age structure of the population also appeared to be an important factor related to the course of the pandemic. The risk of COVID-19 disease severity and death seems to be highly age-related [41,42] as age appears to be a strong risk factor for serious COVID-19 outcomes [41]. While there is no shortage of studies focusing on older people in the available databases of scientific articles, there is a lack of studies that accurately represent the whole population by age categories (classes).

The studies related to the COVID-19 pandemic also address the impact of the pandemic on many aspects of human life. The pandemic has changed the perception of green spaces in cities [43,44,45,46,47,48,49,50], as well as the perception of city public spaces [51,52,53,54]. The pandemic has had, and continues to have, an impact on the real estate market [55].

Worldwide studies have reported that the COVID-19 pandemic has a varying course in different areas in different countries [56,57,58,59,60]. Analyses that rely on generalized values do not always identify problems that arise locally, affect local communities, and require specific actions that do not fit into trends for the country as a whole. In the available literature, it is difficult to find studies that focus on methods for indicating areas with such local anomalies. This article proposes a methodology to identify such situations. To achieve this, the following research objectives are assumed:Analysis of the interrelationships between the three variables that are most commonly used to describe the course of the COVID-19 pandemic worldwide—the number of confirmed COVID-19 deaths, the number of COVID-19 vaccine doses administered, and the number of confirmed SARS-CoV-2 cases.Analysis of the influence of demographic factors described by the age structure and gender breakdown of the population over the course of the COVID-19 pandemic.Identification of the local anomalies over the course of the COVID-19 pandemic.

The first step in the research process was to determine the statistical significance of the analyzed variables using the null hypothesis. This step was accomplished using the spatial autocorrelation tool (Global Moran’s I), by evaluating the *p*-values. Only after rejecting the null hypothesis was it possible to accomplish research objectives #1 and #2, using multiscale geographically weighted regression (MGWR) by comparing R^2^ indices. Next, research objective #3 was accomplished, again using MGWR, but by comparing maps of local R^2^ estimates. The final step of the analysis was to examine the validity of the applied regression model by testing the level of autocorrelation of the residuals in the model, using the spatial autocorrelation tool (Global Moran’s I), and by evaluating z-score indices. To summarize and outline the research process and accompanying research methods, these elements are presented in the graph in Figure 1.

Fulfillment of the above research process and the results obtained from the analyses may be useful for local authorities in developing strategies to further counter the pandemic.

## 2. Materials and Methods

### 2.1. Study Area

Analysis of the interrelation between the number of confirmed SARS-CoV-2 cases, the number of confirmed COVID-19 deaths, and the number of COVID-19 vaccine doses administered, as well as analysis of the influence of demographic factors described by the age structure and gender breakdown of the population over the course of the COVID-19 pandemic, were carried out for the territory of Poland.

In terms of administrative division, the territory of Poland is divided into three levels: voivodships, counties, and municipalities. The first level (‘voivodships’) is the largest in the area and represents large regions; the third level (‘municipalities’) is the smallest in the area and represents towns or rural areas. The second level, the county level, is the most detailed level for which data over the course of the COVID-19 pandemic have been published in Poland: the number of confirmed SARS-CoV-2 cases, the number of confirmed COVID-19 deaths, and the number of COVID-19 vaccine doses administered. For this reason, and because of the high precision of the analysis based on the level of the county, this level was chosen for further study. Poland is divided into 380 counties with an average area of approximately 822 square kilometers (with a standard deviation of 520.55); a high number of analyzed counties is advantageous in this because the chosen regression model requires a large amount of data to achieve the best results [61]. Figure 2 presents the study area—the borders of the counties and voivodships in Poland.

### 2.2. Data Source and Processing

Three variables that are most commonly used to describe the course of the COVID-19 pandemic worldwide were obtained from the database published by Michal Rogalski and Konrad Kalemba [62,63], compiled from reports from Voivodship Sanitary and Epidemiological Stations, County Sanitary and Epidemiological Stations, as well as reports provided by the Polish Ministry of Health. In addition, due to incomplete data on the number of deaths due to SARS-CoV-2 infection, the database was supplemented with archival data, based on the database maintained by Michal Rogalski [64]. These reliable databases are commonly used in many studies that focus on the issue of the COVID-19 pandemic in Poland [65,66,67,68,69,70,71,72,73].

The first vaccine against COVID-19 was administered in Poland at the end of 2020. Initially, medical personnel and others at high risk of exposure to the virus were vaccinated. In 2021, the vaccination program covered all citizens of the country, regardless of age, occupation, and exposure to the virus. The year 2021 was the most challenging year of the COVID-19 pandemic in Poland, due to the high number of cases, as well as the frequent lockdowns [74,75]. For this reason, 2021 was chosen as the study year. In addition, the state of epidemic in Poland was introduced on 20 March 2020, and officially lifted on 13 May 2022; the Polish government justified the lifting of the state of epidemic with the improvement of the epidemiological status in Poland, the reduction in the rapid spread of infections, and the reduction in the number of hospitalized patients. As such, the only full calendar year in which the state of epidemic was in effect was 2021, which further solidifies the choice of this year. The analysis was carried out for data from the period 1 January 2021–31 December 2021. Figure 3 presents a summary of the number of confirmed SARS-CoV-2 cases, the number of confirmed COVID-19 deaths, and the number of COVID-19 vaccine doses administered for the counties of Poland in 2021.

In order to study the influence of the demographic factors described by the age structure of the population over the course of the COVID-19 pandemic, accurate demographic data were obtained for the year 2021, for the counties of Poland. The data come from Statistics Poland, the official government statistical service for Poland. Data from this source are validated and reliable, and are used in a wide range of studies [76,77,78].

Table 1 shows the set of variables used in this article. The first three variables are: the number of confirmed COVID-19 deaths, the number of COVID-19 vaccine doses administered, and the number of confirmed SARS-CoV-2 cases (labeled V1, V2, and V3, respectively). The following 12 variables (labeled V4–V15) describe the age structure of the population of the Polish counties; variable V4 is the total, next V5–V13 are the subsequent age ranges of the population covering the next 10 years (i.e., 0–9 years, 10–19 years, etc.), and the last two (V14 and V15) represents gender breakdown of the population. Table 1 presents an overview of the analyzed variables, with the assigned symbols.

### 2.3. Research Method

The first step of the research was to establish the null hypothesis: the analyzed data are not statistically significant, and the results obtained from the data are the result of random chance. To assess the likelihood of the null hypothesis being true or false, a spatial autocorrelation analysis (Global Moran’s I) was carried out using ArcGIS Pro 3.1 software. Spatial autocorrelation is the presence of systematic spatial variation in a variable, the most common method to measure spatial autocorrelation is to calculate the Moran’s I index [79,80,81,82,83]. The hypothesis test is as follows: if Moran’s I = 0, there is no spatial autocorrelation; if Moran’s I > 0, spatial autocorrelation exists (with positive and negative values indicating positive and negative autocorrelation).

Once the null hypothesis is rejected (i.e., the results of the study are not the result of chance), it was possible to conduct research using geographically weighted regression. This article uses the multiscale geographically weighted regression method (MGWR), the extended and advanced version of GWR. This spatial regression technique explores geographically varying relationships between dependent variables and explanatory variables; MGWR is being used to analyze a variety of problems in geography, urban planning, and various other disciplines [84,85]; studies proved the reliability of MGWR in modeling the COVID-19 issues [86,87,88]. Geographically weighted regression (GWR) allows the coefficients to vary spatially, but all coefficients must change at a similar rate across the study area. The MGWR method is extended by examining relationships at different spatial scales using different bandwidths rather than a single, constant bandwidth. By allowing different bandwidths, MGWR can model a wider range of geographic phenomena than other regression models [86]. Overall, depending on the phenomenon being analyzed, MGWR can provide better estimates, more meaningful interpretations, more accurate predictions, and encounter fewer problems with multicollinearity [89]. The MGWR is expressed as:(1)$$y_{i}=\beta_{0}\left(u_{i},v_{i}\right)+\sum_{k}^{n}\beta_{bwk}\left(u_{i},v_{i}\right)x_{ik}+\varepsilon_{i}$$
where y_i_ is the dependent variable of the i-th, x_ik_ is the explanatory variable of the i-th, u_i_ and v_i_ are the spatial coordinates of the i-th (i.e., the centroid coordinates of the counties), β_bwk_ (u_i_,v_i_) is the estimated coefficient of the k-th explanatory variable for the i-th with the b_w_ bandwidth, and ε_i_ is the residual at the location (u_i_,v_i_).

The MGWR method can be used to estimate the effects of explanatory variables on the dependent variable, and also to identify counties in which the influence of variables differs, to explore and interpret spatial nonstationarity. Due to the complexity of the study objectives, variables V1 and V3 will be treated as both dependent and explanatory variables in subsequent analyses, while the variables V2 and V4–V15 will only be used as explanatory variables. ArcGIS Pro 3.1 software was used to perform the MGWR analysis. GIS software is widely used for spatial analysis in a large variety of scientific fields [90,91,92,93,94].

The degree of influence of the explanatory variables on the dependent variable is described by the R^2^ (or R-squared) index; it is the proportion of the variance of the dependent variable explained by the regression model. By comparing the values of the R^2^ index, it is possible to draw conclusions about the variables.

The R^2^ values present a picture of the analyzed phenomena for the whole of Poland. This is a generalized picture that does not allow an in-depth analysis of such a complex issue as the COVID-19 pandemic. The MGWR regression model allows for the mapping of local R^2^ values, thus exploring spatially varying relationships between variables. Mapping local R^2^ estimates can provide clues about important variables that may be missing from the regression model, i.e., whether the course of the pandemic in Poland may have been influenced by other factors not included in the model.

The final step in the analysis was to examine the level of autocorrelation of the residuals in the regression model. The level of autocorrelation, which indicates the nonrandom nature of the residuals indicates that the regression model is somehow biased by other factors. Spatial autocorrelation analysis (Global Moran’s I) was once again performed using ArcGIS Pro 3.1 software. The explanatory variable explains the dependent variable well only if two conditions are met: a high value of the coefficient of determination (e.g., R^2^ > 0.8), and a weak or insignificant level of spatial autocorrelation in the residuals [95,96].

## 3. Results

### 3.1. Null Hypothesis

Spatial autocorrelation analysis was performed using ArcGIS Pro 3.1 software to assess the probability of the null hypothesis being true or false. The null hypothesis states that the analyzed data are statistically insignificant, and the results obtained from the data are a result of random chance. The spatial autocorrelation (Global Moran’s I) tool measures spatial autocorrelation based on feature locations and feature values, and assesses whether the expressed pattern is clustered, dispersed, or random. Figure 4 presents spatial autocorrelation (Global Moran’s I) reports of the analyzed variables.

The graph displays the z-score values. The graph is bell-shaped, with z-score values between −1.65 and 1.65 indicating that the data are the result of random chance. Values on the left, with a z-score below −1.65, indicate a tendency of the data to be dispersed, and values on the right (z-score > 1.65) indicate a tendency of the data to form clusters.

All analyzed variables are characterized by a tendency to form clusters. A complete summary of the information obtained from the spatial autocorrelation analysis is presented in Table 2 below.

According to the reports, all variables tend to form clusters. Concluding from the *p*-value, all variables are statistically significant (variables V1 and V13 at the 0.05 confidence level, the others at the 0.01 level). The results allowed the rejection of the null hypothesis and to continue the study with the regression model.

### 3.2. Influence Analysis with MGWR

The next step was to analyze the interrelationships between the variables that are most commonly used to describe the course of the COVID-19 pandemic worldwide—the number of confirmed COVID-19 deaths, the number of COVID-19 vaccine doses administered, and the number of confirmed SARS-CoV-2 cases (variables V1, V2, and V3, respectively), as well as to analyze the influence of demographic factors described by the age structure and gender breakdown of the population over the course of the COVID-19 pandemic. The stated research objectives were realized using the MGWR regression model, in ArcGIS Pro 3.1 by Esri, and the results of the R^2^ indices are shown in Table 3 below.

The V2→V1 relation, that is, the influence of the number of COVID-19 vaccine doses administered on the number of registered COVID-19 deaths, shows the lowest R^2^ value among the relations of the first research objective (although the value of 0.94 is very high). The relations: V3→V1, which describes the effect of the number of confirmed SARS-CoV-2 cases on the number of registered deaths as a result of COVID-19, and V2→V3, denoting the effect of the number of COVID-19 vaccine doses administered on the number of confirmed SARS-CoV-2 cases, both showed a value of 0.96. All three analyzed relationships showed very high R^2^ values.

All analyzed variables explain the dependent variables to a very high degree—the lowest R^2^ value is 0.87 (for the V6→V2 relation). This confirms other studies showing population clusters as the most important factor in the development of the COVID-19 pandemic [73,97,98]. Previous research by the authors showed a lesser impact of the younger Polish population on the growth of the pandemic in Poland [73]. The results of the R^2^ index confirm this research by refining it to variable V6, i.e., the population in the 10–19 age range (with a simultaneous low R^2^ for variables V5 and V7, i.e., the 0–9 and 20–29 age ranges).

### 3.3. Local R^2^ Estimates

Using maps of local R^2^ estimates, the third established research objective, i.e., to find local anomalies over the course of the COVID-19 pandemic, was achieved. The voivodships in Poland do not have full autonomy, but possess a certain degree of administrative independence, including decision-making and resource allocation. By superimposing the boundaries of the voivodships on the maps of the local R^2^ estimates, it is possible to additionally examine the influence of the voivodship governments on the analyzed phenomena.

In order to thoroughly examine such a complex phenomenon as the COVID-19 pandemic, the two variables V1 and V3 were treated as both explanatory and dependent variables. The local R^2^ values for Polish counties are shown in Figure 5.

The first relationship V2→V1 indicates to what extent the explanatory variable V2 explains the dependent variable V1. In the case of this relationship, the northwestern part of the Masovian Voivodship, almost the entire area of the Subcarpathian Voivodship, and the eastern part of the West Pomeranian Voivodship can be considered as anomalies, i.e., areas with a local R^2^ value that differs significantly from the R^2^ value for the country as a whole. Analyzing the spatial distribution of counties in specific classes in relation to the overlapping voivodship boundaries, there is a great deal of similarity in the V2→V1 relationship—often voivodship boundaries coincide with class boundaries. For the second V3→V1 relationship, the extent to which the explanatory variable V3 explains the dependent variable V1 was analyzed. The northern part of Masovian Voivodship (primarily Ostrołęka County), the eastern part of West Pomeranian Voivodship, the southwestern part of Lower Silesian Voivodship, as well as the boundary counties of Lesser Poland and Subcarpathian Voivodships can be considered as anomalies, i.e., areas with a local R^2^ value significantly lower than the R^2^ value for the whole country. The last V2→V3 relationship gave very uniform results of local R^2^ values. The exception is the counties of Subcarpathian Voivodship, where there is a cluster of counties assigned to the lowest class of 0.00–0.40, and also none of the counties is in the highest class of 0.91–1.00.

The next step was to analyze deviations and anomalies in terms of local R^2^ values with age ranges. The values of local R^2^ for the counties of Poland, explaining the variable V1 “the number of confirmed COVID-19 deaths”, are shown in Figure 6.

The five population maps for the 10–19 age range and in the 40–79 age range show no clear anomalies in local R^2^ values, nor does the map for the total population. In these maps, the local R^2^ values for counties are mainly contained in two contiguous classes, with minor deviations representing individual counties.

The largest anomalies appear on the map of the 20–29 age range. This is especially noticeable in the north of the Masovian Voivodship (Ostrołęka and Przasnysz Counties), and in the bordering counties of the Lesser Poland and Subcarpathian Voivodships (mainly Gorlice County). Slightly lesser, but also noticeable deviations occur in the eastern part of the West Pomeranian Voivodship, the southeastern part of the Greater Poland Voivodship, and the eastern part of the Masovian Voivodship.

Anomalies of lesser intensity can be observed in other age ranges of the population. The map of the 0–9 population (as well as the map of total female population) shows similar relations to the 20–29. The largest anomaly is Koszalin County, located in the eastern part of the West Pomeranian Voivodeship. Slightly smaller anomalies are found in the eastern part of Greater Poland Voivodship, the northwestern and southern parts of Masovian Voivodship, a large area of Holy Cross Voivodship, and the bordering counties of Lesser Poland and Subcarpathian Voivodship. On the population map 30–39, anomalies can again be seen in the north and west of the Masovian Voivodship, the eastern part of West Pomeranian Voivodship, and the eastern part of Greater Poland Voivodship. In the oldest age range, population over 80, anomalies again appear in the north of Masovian Voivodship, the eastern part of West Pomeranian Voivodship, and the adjacent counties of Lesser Poland and Subcarpathian Voivodship; additionally, anomalies can be seen in the bordering counties of Warmian-Masurian and Podlaskie Voivodships, the western part of Lower Silesian Voivodship, and the central-eastern part of Opole Voivodship.

The values of local R^2^ for the counties of Poland, explaining the variable V2 “the number of COVID-19 vaccine doses administered”, are shown in Figure 7.

Three population maps in the 0–9, 30–39, and 40–49 ranges, as well as the total population show no counties with anomalies. In these maps, the local R^2^ values for the counties are in two adjacent classes.

Again, the largest anomalies appear on the map of the 20–29 age range. Anomalies are clearly visible in the southeastern part of the country (the northern part of the Subcarpathian Voivodship—Jarosław County, Leżajsk County, Lubaczów County, Nisko County, and Przeworsk County; and the eastern part of the Holy Cross Voivodship—Opatów county and Ostrowiec Świętokrzyski County) and on the borders of the Greater Poland and Kuyavian-Pomeranian Voivodships.

There is a clearly visible anomaly in the southeastern part of the country, which is repeatable for the rest of the local R^2^ maps. It covers the entire area of the Subcarpathian Voivodship, as well as individual surrounding counties in the Lublin and Holy Cross Voivodships. In addition, the map of the 10–19 age range shows a large concentration of counties with anomalies, covering the central-northern part of Greater Poland province, along with adjacent counties of Lubusz, West Pomeranian, Pomeranian, and Kuyavian-Pomeranian Voivodships. Gender breakdown maps show almost no anomalies.

The values of local R^2^ for the counties of Poland, explaining the variable V3 “the number of confirmed SARS-CoV-2 cases”, are shown in Figure 8.

The four population maps in the 30–39, 40–49, 70–79, and over 80 ranges do not show any counties with clear anomalies. 

As in the case of the local R^2^ maps for the variables V1 and V2, the largest anomalies are observed in the map of the age range 20–29. This is most evident in the counties on the Subcarpathian and Lesser Poland Voivodship boundaries (primarily Gorlice county). Such a situation also occurs in the north and west of Masovian Voivodship (Ostrołęka County, Mława County, Płock County, and Żuromin County), the eastern part of Masovian Voivodship (Łosice County), the southeastern part of Greater Poland Voivodship, and the southeastern part of West Pomeranian Voivodship (mainly Wałcz County).

Similar relations can be seen on the map of the 10–19 age range, where the areas with the largest anomalies occur in the same locations. Similar conclusions can be drawn for the other maps of the local R^2^ (total population, 0–9, 50–59, 60–69, total female, total male), where anomalies are found in the same counties, only showing lower intensity.

### 3.4. Residual Spatial Autocorrelation

The occurrence of residual spatial autocorrelation was analyzed to assess the performance of the MGWR regression model. The spatial autocorrelation tool (Global Moran’s I) was again used to measure residual spatial autocorrelation. The results are shown in Table 4 below.

Considering the z-scores, the distribution of the residuals does not appear to be significantly different from random. Out of the 39 relations, only 6 values indicating a weak dispersed pattern were registered (z-score ranging from −1.91 to −1.68). These results indicate the random nature of the residuals and confirm the validity of the MGWR regression model.

## 4. Discussion

Previous research by the authors has shown a lesser influence of Poland’s younger population on the spread of the pandemic in Poland [73], which has been confirmed and further detailed in this article by analyzing the interrelation between the three variables that are most commonly used to describe the course of the COVID-19 pandemic worldwide, and analyzing the impact of demographic aspects described by the age structure of the population over the course of the COVID-19 pandemic. This research was conducted using the MGWR regression model; the high R^2^ values obtained confirm other studies showing population clusters as the most important factor in the development of the COVID-19 pandemic.

The “Results” chapter presents the raw results. By taking a comprehensive view and looking at the results in a broader way, a deeper discussion can be made that combines elements of all the variables analyzed. A comprehensive analysis of the studied variables also makes it possible to identify age ranges where the anomalies are repeated. Such a look at the results of the local R^2^ makes it possible to identify the age ranges that deviate most from the picture of the country’s demography. In the case studied in the article, this is noticeable in the 20–29 age range (Figure 6d, Figure 7d and Figure 8d). In this age range, this is particularly evident for the explanatory variables V1 and V3, and also partially confirmed for variable V2. For the dependent variables V1 and V3, areas located in the north of Masovian Voivodship (Ostrołęka and Przasnysz counties), bordering counties of Lesser Poland and Subcarpathian Voivodship (primarily Gorlice county), the eastern part of the West Pomeranian Voivodship (the counties of Koszalin City, Koszalin, Sławno, Szczecinek and Wałcz), the southeastern part of the Greater Poland Voivodship (the counties of Kalisz City, Kalisz, Konin City, Konin, and Pleszew), and the eastern part of the Masovian Voivodship (Łosice County) can all be considered as anomalies. In the case of the V2 dependent variable, other counties with local R^2^ anomalies can be identified. These are located in the southeastern part of the country: the northern part of the Subcarpathian Voivodship: Jarosław County, Leżajsk County, Lubaczów County, Nisko County, and Przeworsk County, and the eastern part of the Holy Cross Voivodship: Opatów County and Ostrowiec Świętokrzyski County. These counties do not stand out in the data input maps (Figure 3), confirming the influence of demographic factors on the explanatory variables. The location of the listed counties is shown in Figure 9. This type of research allows pandemic prevention measures to be targeted to a specific age group in specific areas. Such focused efforts can allow, above all, to optimize the costs of prevention activities.

## 5. Conclusions

The conducted analyses also identify areas with the largest deviations from the picture of the country as a whole in all age groups and for all explanatory variables, as well as areas that can be identified as representative of the country in terms of local R^2^ values. Ostrołęka and Gorlice counties appear most often as anomalies in various explanatory variables and different age ranges; therefore, these counties can be considered as the greatest outliers in the national image. These counties appear to be in need of special, nonstandard interventions. On the other hand, counties for which the local R^2^ values are the smallest deviation from the R^2^ values for the country as a whole can be referred to as representative counties. For the dependent variable V1, these are the counties of Parczew and Lubartów (both counties are located in Lublin Voivodship). For the dependent variable V2, these are the counties of Wałbrzych and Walbrzych City (located in Lower Silesia Voivodeship) and the county of Wieluń (Łódź Voivodship). For the dependent variable V3, a large group of representative counties located in the western part of the West Pomeranian region can be noted. These are the counties of Stargard, Świnoujście, Pyrzyce, Gryfino, Szczecin City, Police, Kamień Pomorski, Myślibórz, and Goleniów. In addition, the representative county of this variable is Kazimierski County, located in Holy Cross Province. Representative counties, due to minimal deviations in R^2^ values, can be taken as a marker or determinant of the pandemic situation nationwide. The selection of representative counties can be useful for assessing the impact of pandemic control activities. The location of the counties discussed is depicted in Figure 10.

The second research objective was to analyze the impact of demographic aspects over the course of the COVID-19 pandemic. The demographic aspects were described by variables V4–V15 representing the structure of the population; the course of the COVID-19 pandemic was described by three variables—the number of confirmed COVID-19 deaths, the number of COVID-19 vaccine doses administered, and the number of confirmed SARS-CoV-2 cases. The analysis using the MGWR method showed the smallest influence of demographic aspects on the V2 variable “the number of COVID-19 vaccine doses administered”. The regression model explains the V2 variable by approx. 93% (R^2^ = 0.93), which may indicate the impact of other variables on the dependent variable. Therefore, it can be concluded that other nondemographic phenomena not included in this study could have influenced the level of vaccination in society. This could be the basis for further research to investigate the influence of cultural, social, or ideological variables over the course of the COVID-19 pandemic.

## Figures and Tables

**Figure 1 ijerph-20-05875-f001:**
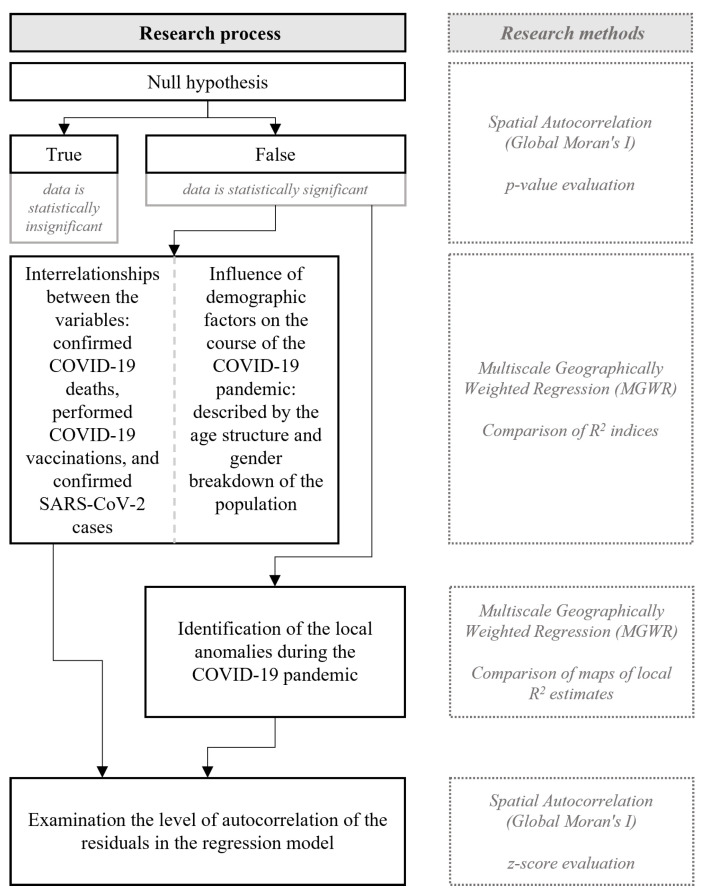
Summary of the research process and research methods. Source: own elaboration.

**Figure 2 ijerph-20-05875-f002:**
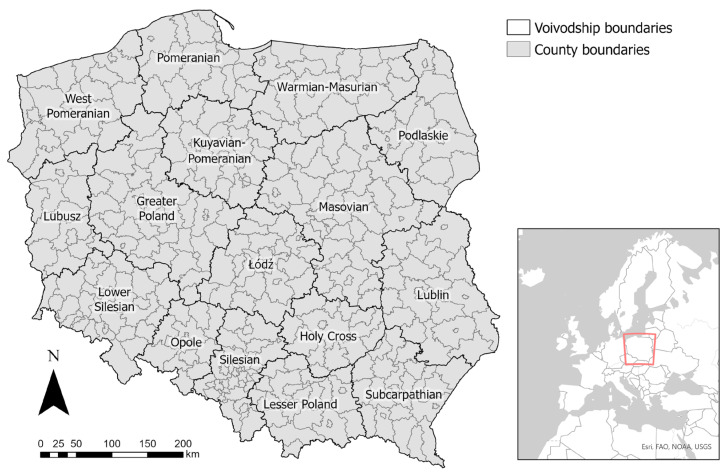
Study area. Source: own elaboration using ArcGIS Pro 3.1 by Esri.

**Figure 3 ijerph-20-05875-f003:**
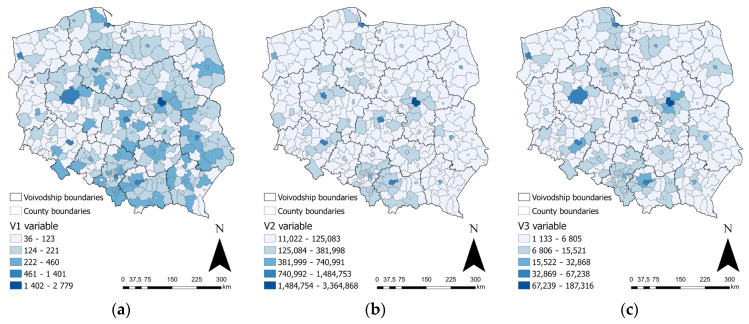
The number of confirmed COVID-19 deaths (**a**), the number of COVID-19 vaccine doses administered (**b**), and the number of confirmed SARS-CoV-2 cases (**c**), in counties of Poland in 2021. Source: own elaboration using ArcGIS Pro 3.1 by Esri, based on [63,64].

**Figure 4 ijerph-20-05875-f004:**
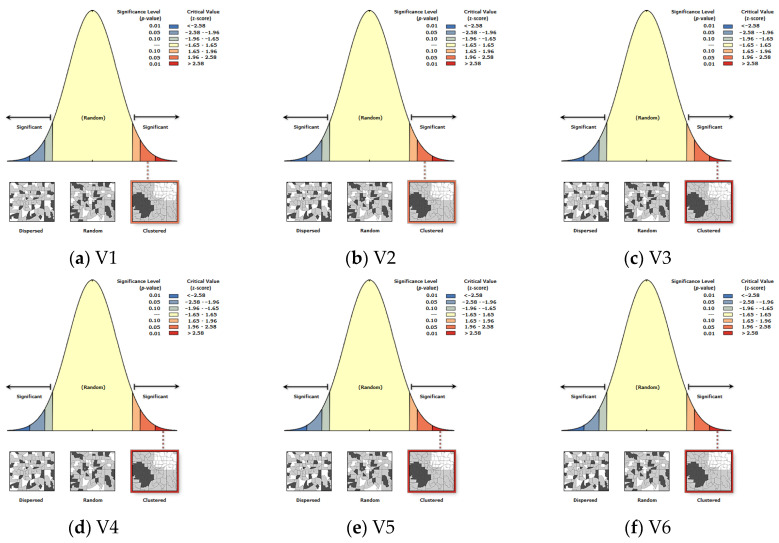
Spatial autocorrelation reports. Source: own elaboration using ArcGIS Pro 3.1 by Esri.

**Figure 5 ijerph-20-05875-f005:**
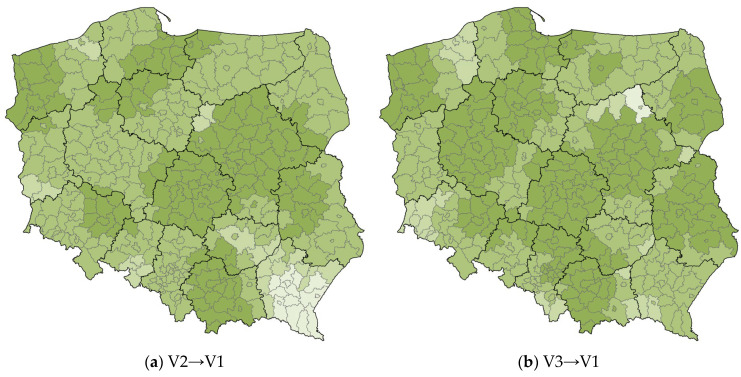
Maps of local R^2^ estimates for the relations: (**a**) V2→V1, (**b**) V3→V1, and (**c**) V2→V3. Source: own elaboration using ArcGIS Pro 3.1 by Esri.

**Figure 6 ijerph-20-05875-f006:**
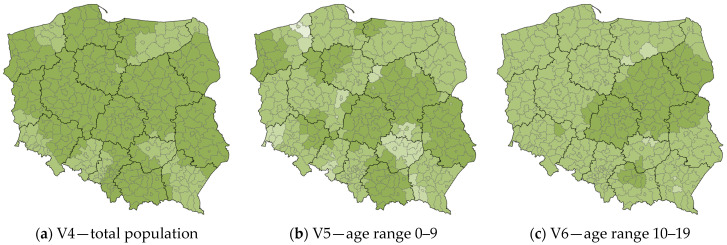
The values of local R^2^ for the counties of Poland, explaining the variable V1 “the number of confirmed COVID-19 deaths”. Source: own elaboration using ArcGIS Pro 3.1 by Esri.

**Figure 7 ijerph-20-05875-f007:**
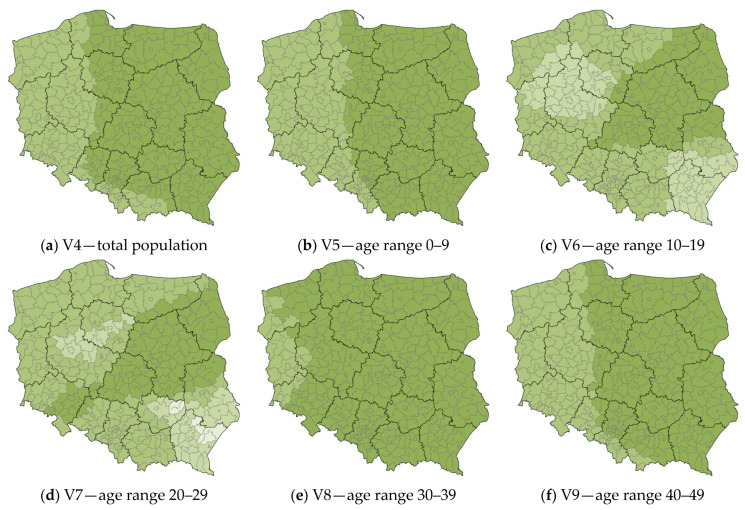
The values of local R^2^ for the counties of Poland, explaining the variable V2 “the number of COVID-19 vaccine doses administered”. Source: own elaboration using ArcGIS Pro 3.1 by Esri.

**Figure 8 ijerph-20-05875-f008:**
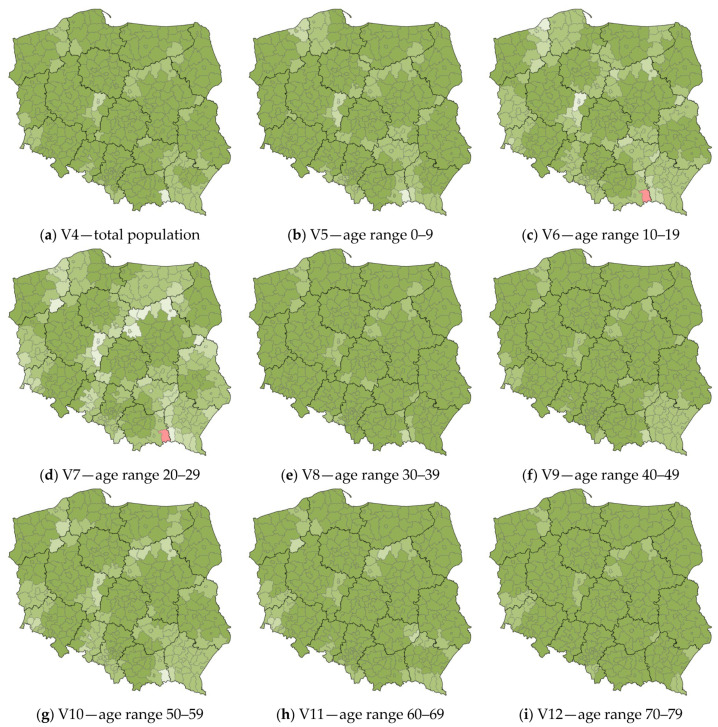
The values of local R^2^ for the counties of Poland, explaining the variable V3 “the number of confirmed SARS-CoV-2 cases”. Source: own elaboration using ArcGIS Pro 3.1 by Esri.

**Figure 9 ijerph-20-05875-f009:**
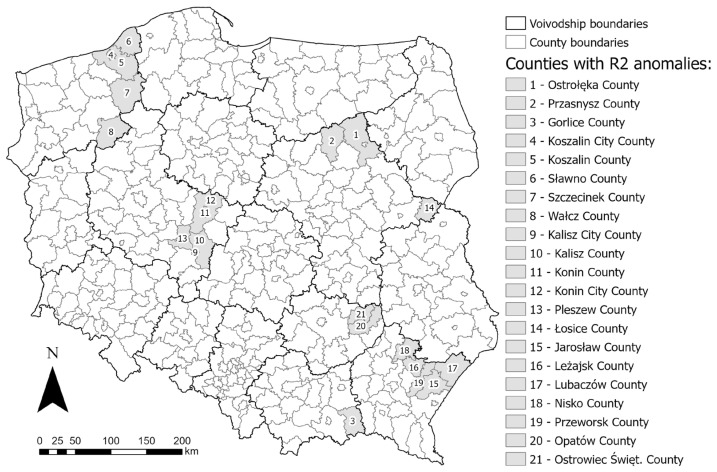
Location of counties where local R^2^ estimates anomalies occur repeatedly. Source: own elaboration using ArcGIS Pro 3.1 by Esri.

**Figure 10 ijerph-20-05875-f010:**
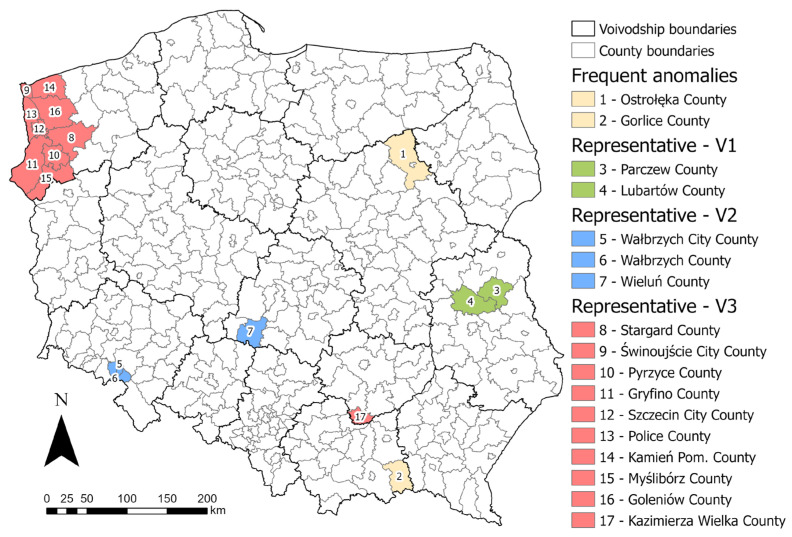
Location of representative counties, and counties that are most often anomalies. Source: own elaboration using ArcGIS Pro 3.1 by Esri.

**Table 1 ijerph-20-05875-t001:** Overview of the analyzed variables with the assigned symbols. Source: own elaboration.

Variable	Symbol
The number of confirmed COVID-19 deaths	V1
The number of COVID-19 vaccine doses administered	V2
The number of confirmed SARS-CoV-2 cases	V3
Population—total population	V4
Population—age range 0–9	V5
Population—age range 10–19	V6
Population—age range 20–29	V7
Population—age range 30–39	V8
Population—age range 40–49	V9
Population—age range 50–59	V10
Population—age range 60–69	V11
Population—age range 70–79	V12
Population—age range over 80	V13
Population—total female	V14
Population—total male	V15

**Table 2 ijerph-20-05875-t002:** Summary of the spatial autocorrelation. Source: own elaboration on the basis of ArcGIS Pro 3.1 by Esri.

Symbol	*p*-Value	Z-Score	Moran’s I index	Spatial Pattern	Confidence Level
V1	0.02	2.40	0.07	Clustered	5%
V2	0.01	2.51	0.06	Clustered	1%
V3	0.00	5.97	0.20	Clustered	1%
V4	0.00	4.11	0.12	Clustered	1%
V5	0.00	5.44	0.16	Clustered	1%
V6	0.00	5.71	0.17	Clustered	1%
V7	0.00	4.48	0.14	Clustered	1%
V8	0.00	3.89	0.11	Clustered	1%
V9	0.00	4.69	0.14	Clustered	1%
V10	0.00	4.20	0.13	Clustered	1%
V11	0.00	3.04	0.09	Clustered	1%
V12	0.01	2.63	0.08	Clustered	1%
V13	0.04	2.03	0.07	Clustered	5%
V14	0.00	3.96	0.12	Clustered	1%
V15	0.00	4.27	0.13	Clustered	1%

**Table 3 ijerph-20-05875-t003:** Results of MGWR analysis—values of R^2^ indices for analyzed variables. Source: own elaboration on the basis of ArcGIS Pro 3.1 by Esri.

Symbol	MGWR R^2^ Value		
	V1	V2	V3
V1 *	--	--	--
V2 **	0.94	--	0.96
V3 **	0.96	--	--
V4 **	0.97	0.93	0.99
V5 **	0.95	0.91	0.99
V6 **	0.93	0.87	0.98
V7 **	0.94	0.89	0.97
V8 **	0.97	0.94	0.99
V9 **	0.96	0.92	0.99
V10 **	0.96	0.91	0.99
V11 **	0.98	0.95	0.99
V12 **	0.98	0.97	0.99
V13 *	0.97	0.97	0.98
V14 **	0.97	0.95	0.99
V15 **	0.96	0.93	0.99

Note: **—statistically significant at the *p* < 0.01 level, *—statistically significant at the *p* < 0.05 level.

**Table 4 ijerph-20-05875-t004:** Spatial autocorrelation results for MGWR residuals, values of z-score. Source: own elaboration on the basis of ArcGIS Pro 3.1 by Esri.

	V2	V3	V4	V5	V6	V7	V8	V9	V10	V11	V12	V13	V14	V15
V1	−1.51	−1.75	−1.91	−0.49	−0.74	−1.54	−0.52	−1.31	−1.23	−0.72	0.06	−0.43	−1.45	−1.00
V2			−1.71	−0.51	−0.61	−0.93	−1.17	−1.58	−1.68	−1.37	−1.21	−1.37	−1.62	0.16
V3	−1.88		−0.65	−1.59	−1.35	−1.38	−1.18	−1.08	−1.06	−0.99	−1.34	−1.51	−0.95	−1.91

## Data Availability

Not applicable.

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
