# Peer review of "Multiscale Geographically Weighted Regression in the Investigation of Local COVID-19 Anomalies Based on Population Age Structure in Poland"

_ijerph, 2023, doi:10.3390/ijerph20105875_

Round 1

Reviewer 1 Report

The author attempted to explore the relationship between COVID-19 and population structure using MGWR to assist in the prevention and control of COVID-19, which is a somewhat novel study. Overall, I think the topic discussed in this article is interesting and has some practical value. However, before publication, I still believe that there are some issues that can be corrected.

1. The statements in the introduction are somewhat tedious, and there is no strong logical organization around a core issue. This type of statement makes it difficult to focus on the research and the scientific problem is not apparent. For example:

(1) Lines 80-84, the purpose of this statement is unclear. If it is to express the impact of COVID-19, it is not highly relevant to the topic.

(2) Lines 85-98 involve three research objectives. However, I believe that research objectives should focus on a core issue rather than being all-encompassing, which is more like a statement of research content.

(3) In particular, the second research objective (Lines 96-97) is difficult to understand. The article cited references [47-50] to indicate that age has an important impact on the prevalence of COVID-19, so why does this article still need to study the "relationship between population structure and COVID-19"? Isn't this duplicative research? 

2. The statements in the research methods section lack hierarchy and a clear logical structure, and are more about describing the research process of this article. I believe that the research process should be distinguished from the research methods and clearly explained. The research methods section should only describe the specific methods or models applied to achieve a certain research purpose. For example:

(1) Lines 174-178, the sudden introduction of data overview in the research methods can interrupt the reader's thinking about the stated methods.

(2) Line 185, the expression "Moran's I → 0" is not standardized, does it describe the trend of Moran's index value clearly?

(3) Lines 210-212, why do you suddenly explain the data variables here? It is very abrupt. 

3. Line249: The multi-scale geographically weighted regression (MGWR) model aims to address the problem of the traditional geographically weighted regression model, which assumes that the effect of all explanatory variables on the response variable is uniform (i.e., uniform bandwidth), resulting in fitting results that do not match the actual prediction effect. Compared with the uniform bandwidth of traditional geographically weighted regression, the MGWR model allows each explanatory variable to have a different effect scale on the response variable, that is, different bandwidths (because some independent variables have higher local variation characteristics, it is more suitable to choose a narrow bandwidth for regression calculation, while some independent variables are relatively stable globally, so it is suitable to choose a larger bandwidth for regression calculation). This is to detect the differences in the way and scale in which explanatory variables affect the dependent variable in different geographic locations, that is, spatial heterogeneity. For example, due to different levels of regional autonomy, the scale of the COVID-19 prevention and control policies may be at the county level, while the scale of the COVID-19 vaccination policy may be at the city or even national level. Therefore, its effect scale is larger because the vaccination policy may be unified across the country. The results of this paper mainly describe the R2 of the model fitting between different explanatory variables and the dependent variable, and I think R2 only reflects the explanatory power of the explanatory variables on the dependent variable, that is, the fitting effect of the model. The description of the results in the paper does not reflect the main features of the MGWR model, that is, according to the bandwidths of different explanatory variables and combined with the regression coefficients of each factor and R2, to explain their spatial scale differences and spatial effects on the dependent variable.

 4. Line241, 277, 415: "Error! Reference source not found" appears several times. Attention should be paid to formatting details.

 5. Line243: In Table 2, what does "C" represent? This needs to be explained.

 6. Line250-277: This part over-describes the experimental process rather than the results. The results section should be concise and straightforward in describing the experimental results.

 7. Conclusion and discussion should be separated. The current way of stating is more about results and discussion, and the conclusion part needs to be reformed into a brief paragraph to summarize the important research findings.

Author Response

We sincerely thank You for taking the time to review our article. Thank You for Your valuable suggestions to the manuscript, that helped us improve the paper and the approach of the study. We tried to address all of Your suggestions. Please see attached a step-by-step response to Your suggestions and comments. We hope that the improvements we have made will fully satisfy You and meet with Your positive evaluation.

Sincerely,

Authors.

Reviewer 2 Report

The manuscript has employed Multiscale Geographically Weighted Regression to test COVID-19 relation with population age structure in Poland. It is an interesting topic, however, it needs extensive revision. 

There are many other studies in Europe and outside which assessed the effects of demographic indices on COVID-19 spread and cluster formation. I suggest authors to review them.

The introduction section starts with very basic information about COVID-19. Which are not necessary at all.

The citation of other studies in the text are inappropriate. For example: Line 50- A very important part of research, undertaken by various scientists, has become the issue of COVID-19 vaccine.................... This is not the proper way to write.

The methods section is not properly written. Please avoid unnecessary explanations on already established methods. It needs to reorganize and rewrite. 

There are figures and tables which could be moved to the supplement. Please keep major results only in the main paper. 

Find some other comments in the PDF.  

The manuscript needs extensive revision. 

Author Response

(The authors gave the same response as above.)

Reviewer 3 Report

This manuscript examines some aspects of the COVID-19 pandemic for the area of Poland. The authors analyze the interrelationships between the number of confirmed SARS-CoV-2 cases, the number of confirmed COVID -19 deaths, and the number of COVID-19 vaccinations performed. The multiscale geographically weighted regression method is used. The influence of demographic factors described by the age structure of the population is also studied. The study may be of interest to decision makers in countering further waves of the pandemic. In my opinion, the manuscript needs a major revision, taking into account the following comments:

1. The definitions of variables, V1, V2, and V3 mentioned on page 8 are different from those in Table 1, and consequently the interpretations of the relations, V2-->V1, V2-->V3, and V3-->V1 become confusing.

2. I think it is necessary to investigate the influence of gender. Therefore, I suggest to consider the breakdown by gender at least in the total population, V4.

3. The entire manuscript contains unwarranted repetition of definitions of variables and relations.

4. In the Discussion and Conclusions section, it is recommended to address the limitations of the study. For example, how meaningful are the results in the case of mutation and the generation of new strains?

5. In Figure 3, it is not necessary to repeat the same figure for all variables.

6. There are many repeated sentences throughout the manuscript.

7. There are errors in references to some tables in the text.

There are many linguistic and stylistic errors that need to be corrected

Author Response

(The authors gave the same response as above.)

Reviewer 4 Report

The manuscript “Multiscale Geographically Weighted Regression in the investigation of local COVID-19 anomalies based on population age structure in Poland” is excellent piece of write up with excellent research question, appropriate methods and well-illustrated  results.   I recommend to accept the manuscript in the current form. However, I strongly suggest to have a English check before the publication.

moderate English  editing is necessary 

Author Response

Response to Reviewer 4

We sincerely thank You for taking the time to review our article. Thank You for Your valuable suggestion to the manuscript, that helped us improve the paper. We hope that the improvements we have made will fully satisfy You and meet with Your positive evaluation.

The manuscript “Multiscale Geographically Weighted Regression in the investigation of local COVID-19 anomalies based on population age structure in Poland” is excellent piece of write up with excellent research question, appropriate methods and well-illustrated results. I recommend to accept the manuscript in the current form. However, I strongly suggest to have a English check before the publication.

Moderate English editing is necessary 

Thank You for the positive review, the article has gone through linguistic correction.

Sincerely,

Authors.

Reviewer 5 Report

This manuscript paper conducts research on COVID-19, using the Multiscale Geographically Weighted Regression, an analysis of the interrelationships between the number of confirmed SARS-CoV-2 cases, the number of con- firmed COVID-19 deaths, and the number of performed COVID-19 vaccinations were conducted. This study has a clear research objective, however, at present, there are major revisions before this paper can be published in this journal, the authors should further modify them.

1.       The methods for COVID-19 anomalies based on population age structure should be further summarized in the section of introduction?

2.       In terms of methodology, the authors need to supplement the overall technical roadmap of this study. Moreover, the method description needs to be closely integrated with the research task.

3.       In Figure 3, the subgraphs look almost identical, the author could add some annotations for readers to better view and compare.

4.       How to prove the effectiveness of the method proposed in this article? How does it improve compared to other methods?

5.       In Lines 169~170, Lines 241~242, Line 277, the incorrect citation mark has occurred “Error! Reference source not found”, please check the whole paper.

Author Response

(The authors gave the same response as above.)

Round 2

Reviewer 2 Report

Thank you for revising the manuscript. 

Regarding the justification for keeping the basic info about COVID-19 at the beginning of the Introduction, I am not convinced by your justification. Readers have access to many publications on COVID-19 from previous years which have already included the basic information about its cause, outbreak, etc. Please think of the word economy in writing. 

Minor English editing is required. 

Author Response

Thank You very much for Your review, Your comments allowed us to expand and improve the article, and Your helpful comments certainly enhanced the quality of the paper.

We have followed Your comment, the reader does not need such a basic introduction to the COVID-19 pandemic. We agree with Your suggestion, the paragraph has been removed from the article.

The article has undergone an additional linguistic correction. We sincerely thank You for taking the time to review our article. Thank You for Your valuable suggestion on the manuscript, which helped us to improve the paper. Following Your comments has significantly improved the quality of the article.

Regards,

Authors

Reviewer 3 Report

I believe that the manuscript has been sufficiently improved. I recommend the manuscript for publication in IJERPH.

Author Response

We sincerely thank You for taking the time to review our article. Thank You for Your valuable suggestion to the manuscript, that helped us improve the paper. Following Your comments has significantly improved the quality of the article.

Regards,

Authors

Reviewer 5 Report

My concerns have been well replied, this paper can be accepted now.

Author Response

(The authors gave the same response as above.)
